# Protective Effect of Olive Oil Microconstituents in Atherosclerosis: Emphasis on PAF Implicated Atherosclerosis Theory

**DOI:** 10.3390/biom13040700

**Published:** 2023-04-20

**Authors:** Smaragdi Antonopoulou, Constantinos A. Demopoulos

**Affiliations:** 1Laboratory of Biology, Biochemistry and Microbiology, Department of Nutrition-Dietetics, School of Health Sciences and Education, Harokopio University, 17676 Athens, Greece; 2Laboratory of Biochemistry, Faculty of Chemistry, National & Kapodistrian University of Athens, 15784 Athens, Greece; demopoulos@chem.uoa.gr

**Keywords:** atherosclerosis, cardiovascular diseases, olive oil, olive oil by-products, PAF, phenolic compounds, vitamin E, platelets, sterols

## Abstract

Atherosclerosis is a progressive vascular multifactorial process. The mechanisms underlining the initiating event of atheromatous plaque formation are inflammation and oxidation. Among the modifiable risk factors for cardiovascular diseases, diet and especially the Mediterranean diet (MedDiet), has been widely recognized as one of the healthiest dietary patterns. Olive oil (OO), the main source of the fatty components of the MedDiet is superior to the other “Mono-unsaturated fatty acids containing oils” due to the existence of specific microconstituents. In this review, the effects of OO microconstituents in atherosclerosis, based on data from in vitro and in vivo studies with special attention on their inhibitory activity against PAF (Platelet-Activating Factor) actions, are presented and critically discussed. In conclusion, we propose that the anti-atherogenic effect of OO is attributed to the synergistic action of its microconstituents, mainly polar lipids that act as PAF inhibitors, specific polyphenols and α-tocopherol that also exert anti-PAF activity. This beneficial effect, also mediated through anti-PAF action, can occur from microconstituents extracted from olive pomace, a toxic by-product of the OO production process that constitutes a significant ecological problem. Daily intake of moderate amounts of OO consumed in the context of a balanced diet is significant for healthy adults.

## 1. Introduction

Cardiovascular diseases (CVDs) remain the leading cause of mortality worldwide, with ischemic heart disease and stroke being the two top causes of death in 2019 [1,2]. Undoubtedly, the COVID-19 pandemic will have a serious impact on the above statistics and future research will show if CVDs and COVID-19 share common pathophysiological pathways.

The first stage of CVDs beside others, is mainly atherogenesis, which is characterised by the formation of fatty lesions in the artery wall, with the end result of atheromatous plaque formation. The term atherogenesis comes from the ancient Greek word athero-αθήρωμα (sack/bag) and genesis-γένεση (generation/formation). Even though a significant number of papers on atherogenesis have appeared over the last decades, the mechanisms that have been developed to justify the initiating event of atheromatous plaque formation and still receiving the widest acceptance, are the theories of inflammation, of oxidation, and the response-to-retention theory while none of these theories alone can fully explain the phenomenon. According to the inflammation theory that was proposed by Russell Ross [3], atherogenesis initiates from a local inflammation inside the artery. In the theory of inflammation, the cells that play a key role are the endothelial cells by becoming dysfunctional and with increased permeability, platelets that are activated and form thrombus, monocytes that initially become macrophages and then by up taking cholesterol are converted to foam cells and lastly the smooth muscle cells that proliferate and migrate, creating the fibrous cap of the atheromatous plaques. The theory of oxidation, or oxidative modification, was introduced by Brown and Goldstein [4] and attributes the initiation of atherogenesis to the oxidation of mainly LDL and the consequent generation of oxidized-LDL (Ox-LDL). The disturbance of balance between reactive oxygen and/or nitrogen species formation (ROS, RNS) and the radical scavenging systems leads to excessive ROS/RNS generation and oxidation of LDL particles with atherogenic properties [5]. During this process, LDL phospholipids are also oxidized and oxidized phospholipids (OxPLs) with PAF-like activity as well as Platelet-Activating Factor (PAF) are generated [6]. According to the response-to-retention theory that complements and reinforces the two previous theories, subendothelial retention of apolipoprotein (apo)B–containing lipoproteins bound by the extracellular material have an increased susceptibility to chemical modifications, including oxidation and foam cells formation [7].

In addition, our research team proposed in 2003 that PAF, a potent lipid inflammatory mediator, could be implicated in the initiation and propagation of atherosclerosis and developed a new theory “*The PAF implicated atherosclerosis theory*” [8]. This theory was published as the main article in the Journal of Lipid Sciences and Technology (official journal of the European Federation of Fat and Lipids Science and Technology, Euro Fed Lipid) with reference to the cover. PAF identified as 1-O-alkyl-2-acetyl-*sn*-glycero-3-phosphocholine [9] is an analogue of phosphatidylcholine with a long aliphatic chain with an ether bond in the *sn*-1 position of the glycerol backbone that varies, with the 16:0 moiety being the most prominent and potent, and an acetyl group (instead of fatty acid) in the *sn*-2 position that is responsible for its characteristic biological action (Figure 1). The term PAF was applied in 1972 by Benveniste, Henson, and Cochrane [10] in order to describe the activities of PAF that were known until then, namely platelet activation and aggregation. PAF has now been recognized as a primitive lipid mediator with several actions in almost all systems of humans. A recent review provides a historical account of the discovery of PAF and its research in physiology and pathophysiology [11].

Among the modifiable risk factors for CVDs, diet plays a significant role. The Mediterranean diet (MedDiet), among other healthy dietary patterns, has been recently evaluated as the most beneficial one for cardiometabolic risk factors, for preventing cardiovascular recurrence even without changes in cholesterol levels or weight and is also associated with low CVDs mortality [12,13]. Even though geographical differences exist in the traditional MedDiet, they all include OO as the main source of the fatty components of the MedDiet, and consequently, a great deal of research has been focused on the effect of OO on atherosclerosis. The superiority of OO over the other “Mono-unsaturated fatty acids (MUFA) containing oils” in the protection against the development of atherosclerosis has been attributed to the existence of specific microconstituents. Some of these microconstituents are present in small amounts in the other seed oils as well, but they are removed during the refining process required in all seed oils, with the exception of sesame oil, which retains these trace elements.

Therefore, the scope of this review is to present the OO microconstituents and their protective effects in atherogenesis, based on data from in vitro experiments, from animal models and from humans with special attention on their inhibitory activity against well-known PAF actions. We also suggest that the anti-atherogenic effect of OO is attributed to the synergistic action of its microconstituents, mainly polar lipids that act as PAF inhibitors, tyrosol and oleuropein as well as α-tocopherol that also exert anti-PAF activity. This beneficial effect, also mediated through anti-PAF action, can additionally occur from microconstituents extracted from olive pomace, a toxic by-product of the OO production process that constitutes a significant ecological problem.

## 2. The PAF Implicated Atherosclerosis Theory

The proposed theory links the recognized biological actions of PAF to the current theories of atherosclerosis, combining our experimental data with the existing ones of the international research. In addition, it proposes an explanation of the beneficial role of the MedDiet against atherosclerosis and cardiovascular diseases, in a biochemical approach.

PAF is the most potent lipid inflammatory and thrombotic factor. A final concentration of 10^−10^ M PAF in vivo causes anaphylactic shock and 10^−8^ M causes death, in rabbits [14]. Data from in vitro studies reveal that 10^−5^ µg of PAF is required to induce washed rabbit platelet (WRP) aggregation along with 50% platelet secretion of serotonin, while 10^−2^ µg of adenosine diphosphate (ADP) and the same amount of thrombin (THR) are required to induce the same effect, being 1000 times less active than PAF. Additionally, 4 µg of arachidonic acid (AA) (100,000 times less active) and 10 µg of collagen (COL) are required to cause WRP aggregation (1,000,000 times less active) [15,16]. Activated platelets, apart from their significant role in thrombosis and hemostasis, induce cell–cell interactions, including endothelium, and are recognized as cellular modulators of both the innate and adaptive immune system [17,18]. Additionally, it is recently reported that platelet incubation with the Spike protein, either in human platelet-rich plasma (PRP) or in WRP, results in augmentation of PAF-induced aggregation [19]. For all these reasons, the reduction in platelet aggregation, i.e., the reduction of the ability of platelets to aggregate due to the various aggregating factors, has been accepted by the European Food Safety Authority (EFSA) to contribute to overall vascular and cardiovascular health [20].

PAF is biosynthesized by many cell types under different stimuli and presents a variety of biological responses acting in both an autocrine and paracrine way by binding to the G-protein-coupled receptor (PAF-R) located on the plasma and nuclear membrane (Figure 2). The produced PAF can either remain intracellular, thereby acting as an intracellular mediator or is exposed on the cell membrane, thereby acting as a transcellular mediator. Its signal transduction pathway through G-proteins is complex and cell-dependent [21]. In most cases, PAF binding to PAF-R leads to a pro-inflammatory phenotype, while activation of the PAF-R in macrophages and in dendritic cells is associated with a suppressor phenotype [22,23,24]. It should be noticed that OxPLs also bind to PAF-R and exert PAF-like activity [25]. It has also been reported that the PAF-PAF-R complex interacts with and activates Toll-Like Receptors 4 (TLR4) and there is evidence that this also occurs with TLR2 [26].

PAF levels in cells, tissues, and biological fluids are mainly regulated via its enzymatic biosynthesis and catabolism [27]. PAF is biosynthesized by two distinct pathways, the de novo and the remodelling pathway. The de novo pathway is responsible for the constitutive production of PAF with 1-alkyl-2-acetyl-*sn*-glycerol cholinephosphotransferase (PAF-CPT) catalysing the final reaction as a key enzyme [28], while the remodelling pathway starts with the hydrolysis of long chain fatty acids in the *sn*-2 position of glyceryl-ether analogues, mostly of AA, followed by acetylation with the action of the acetyl-CoA:lyso–platelet-activating factor acetyltransferases (Lyso-PAF-AT). Oxidative stress is reported to induce noticeable activation of Lyso-PAF-AT [29]. The isoform of this enzyme (LPCAT2) is activated under inflammatory conditions [30]. Degradation and inactivation of PAF occurs by hydrolysis of the acetyl group in the *sn*-2 position by intracellular PAF-specific acetylhydrolase (PAF-AH) and its plasma isoform lipoprotein-associated phospholipase A_2_ (LpPLA_2_) [31]. In addition, PAF can be formed non-enzymatically during the oxidative modification of LDL, which results in increased PAF levels and in the production of OxPLs with PAF-like activity, while the activity of Lp-PLA_2_ decreased [32]. Dietary antioxidant capacity as well as healthy dietary patterns rich in OO, legumes, and vegetables were associated with favourable changes of both PAF levels and the activity of its metabolic enzymes [33,34]. Additionally, PAF action can be modulated by the presence of its inhibitors that can either be endogenously synthesized [35] or are provided by food intake, such as micronutrients present in the characteristic foods of the MedDiet [36]. Finally, PAF levels can also be modified by the action of the PAF-dependent transacetylase (TA), which transfers the acetyl group of PAF to synthesize either acyl-analogs of PAF or C2- ceramide [29].

From all the above, PAF is an important link in the initiation and propagation of atherosclerosis and according to “*the PAF implicated atherosclerosis theory*” [8], the uncontrolled PAF production by the disruption of the aforementioned pathways under pathological conditions or from LDL oxidation, activates cells, such as platelets, neutrophils, monocytes, etc., thus initiating a rapid, local inflammatory response in the vessel, which leads to a dysfunctional endothelium with increased permeability. This process permits blood cells and ox-LDL to move into sub-endothelial space, resulting in foam cells formation as well as proliferation of smooth muscle cells [37]. Consequently, foam cells, platelets, smooth muscle cells as well as lipids, such as cholesterol, adhere and form the complex mixture of cells and lipids that composes the atheromatic plaque (Figure 3). In addition, PAF, apart from its implication in atherogenesis, has also direct actions in heart functions leading in CVDs [11].

## 3. Olive Oil

From several epidemiological and clinical studies, it has been documented that OO, and especially virgin olive oil (VOO) intake, has many beneficial effects in atherosclerosis and in the majority of CVDs. Briefly, the intake of OO results in reduced blood pressure, improved blood lipid metabolism, by reducing triacylglycerols and LDL oxidation, while increasing HDL levels, improved plasma glucose, and HbA1c as well as post-prandial glucose levels and contributes to an enhanced endothelial function. OO exerts anti-inflammatory, anti-oxidant, and anti-thrombotic actions [38]. Recent meta-analyses and review articles suggest that OO microconstituents are mainly responsible for its antiatherogenic properties, either alone or in synergy [39,40]. Therefore, this review focuses on data from in vitro experiments, from studies conducted with animal models as well as from human studies that were carried out mainly with discrete and separated OO microconstituents in an attempt to highlight those that play a determinant role in modulating atherosclerosis development.

The basic composition of OOs is presented in Figure 4. It varies depending on the variety of olive tree, the agronomic conditions, e.g., soil and climate, the harvesting method, and the OO production system. OO consists of approximately 98% neutral lipids (NL), with the main component being triglycerides (TG), and 1–2% of polar lipids (PL) that contain a great variety of complicated chemical compounds. The terms saponifiable and nonsaponifiable fractions that are frequently used in oils generally are not identical in terms of composition with the fractions of NL and PL, respectively, since phytosterols, hydrocarbons, squalene, α-, β-, γ-tocopherols, and triterpenes that are present in the NL fraction are extracted in the nonsaponifiable fraction.

Among the microconstituents that belong to the class of NL, mono-unsaturated fatty acids (MUFAs), and especially oleic acid which is the main component of OO (60–83% of the total fatty acids), have attracted attention in relation to their beneficial effects. Other MUFAs are also present in smaller quantities (Figure 4). Extra virgin olive oil (EVOO) also contains significant amounts of poly-unsaturated fatty acids (PUFAs), with linoleic acid (3–19%) being the main one. Other microconstituents present in the NL are squalene (1–6 g/kg), phytosterols (0.8–2.6 g/kg), and vitamin E (with α-tocopherol as the predominant form, <0.2 g/Kg). The triterpenes, erythrodiol, uvaol, oleanolic, and maslinic acids, are found in very low concentrations in VOOs but are present in detectable amounts in olive pomace oil. PL fraction includes phenolic compounds (50–1500 mg/kg) that are classified into four main groups: (a) phenols (100–300 mg/Kg), tyrosol (T) and hydroxy-tyrosol (HT) and phenolic acids e.g., p-coumaric, vanillic; (b) lignans (1–100 mg/kg), e.g., pinoresinol; (c) flavonoids (luteolin and apigenin, 1–7.5 mg/kg); and (d) secoiridoids (4–5 mg/kg), oleuropein. Among the above compounds, HT is the most abundant one. The above amounts of phenolic compounds are present in VOO or EVOO. In a recent paper, a significant number of phospholipids are identified in EVOO, with phosphatidic acids and phosphatidylglycerols being the most abundant species [41]. Glyceryl-ether glycolipids are also included in the PL fraction and a representative structure is shown in Figure 4 [42]. In order to preserve the fraction of PL that includes phenolic compounds and glyceryl-ether glycolipids, oils must not be processed or refined.

### 3.1. In Vitro Experiments

The majority of the in vitro studies with OO microconstituents has been performed by estimating the ability of microconstituents to inhibit platelet aggregation induced by various agonists, due to the predominant role of platelets in atherosclerosis. In addition, the inhibition of PAF-induced platelet aggregation is also an internationally accepted technique to confirm the existence of PAF inhibitors. In these studies, WRP, PRP, gel-filtered platelets, or whole blood (WB) from healthy human donors were used as the source of platelets. Briefly, the platelets are stimulated to aggregate by the addition of a specific concentration of various agonists, such as ADP, THR, AA, COL, and PAF, in an aggregometer cuvette at 37 °C with stirring, and the corresponding aggregation curve is recorded. Consequently, the same experiments are performed with the addition of various concentrations of the specific OO microconstituent—usually with up to 10 min preincubation—before the addition of the aggregating agents and the corresponding aggregation curve is also recorded. The inhibitory capacity of a compound is expressed as IC_50_ (Inhibitory Concentration for 50% Inhibition) which is the amount of the compound required to cause 50% inhibition to the aggregation induced by the agonists. The existing data display a range in IC_50_ values for each of the OO microconstituents tested, since the results depend on the experimental conditions, such as the source of platelets and the agonist used.

The main microconstituent from the NL fraction is the oleic acid, which was reported to inhibit platelet aggregation induced by ADP, AA, COL, and PAF, with IC_50_ values ranging from 30 to 100 µM [43,44,45]. High concentrations of vitamin E, specifically 2.0 mM of α-tocopherol (αT) and 0.5 mM of γ-tocopherol (γT), were needed to sufficiently inhibit platelet aggregation induced by ADP, COL, AA, and epinephrine and to completely inhibit platelet release, even though vitamin E deficiency has been associated with platelet hyperaggregability [46,47]. Aspirin-treated platelets incubated with 50–100 μM vitamin E inhibited platelet aggregation induced by COL and PAF, and also αT (20–200 μg/mL) inhibited PAF-induced platelet aggregation in WB while it did not inhibit PAF-induced aggregation in human PRP [48,49]. The above data led the authors to suggest that the effect of vitamin E in the case of WB should be upon PAF synthesis and release by other cells and is unrelated to ADP or thromboxane A_2_.

From the fraction of PL, several phenolic compounds are able to inhibit in vitro platelet aggregation and a recent review summarizes these effects [50]. Among the various phenolic compounds, the phenol HT is the most potent one. HT caused inhibition with concentrations at the order of μM, with IC_50_ values ranging from 23–738 μM against ADP, from 67–195 μM against COL, and its IC_50_ value against AA was 195 μM. The higher IC_50_ values were observed when WB was used [51,52,53]. Taking under consideration that VOOs contain 100–300 mg HT/Kg, the above IC_50_ values correspond to more than 500 μL of OO. The other phenol, T inhibited PAF-induced WRP aggregation with IC_50_ value at 2.2 mM [54]. Lastly, oleuropein showed an inhibitory effect against the PRP aggregation induced by PAF with IC_50_ at 0.41 mM, by ADP with IC_50_ at 4.4 mM and by AA with IC_50_ at 0.95 mM [51,55]. In relation to platelet function, HT (63–400 μM) inhibited thromboxane B_2_ (TxB_2_) production induced by various agonists as well as the platelet secretion of platelet-derived growth factor and soluble CD40L. The last effect was also observed with oleuropein at 500 μM [51,52,56]. In addition, HT (IC_50_ at 15 mM), oleuropein (IC_50_ at 80 mM), and T (IC_50_ at 500 mM) were reported to inhibit leukotriene B_4_ (LTB_4_) production in platelets and leukocytes [57].

Finally, our group has demonstrated that apart from the inhibitors of PAF- and THR- induced aggregation present in OO PL fraction, weak PAF-like agonists also exist in the same fraction. These agonists, chemically identified as glyceryl-ether glycolipids, bind to PAF-R and induce aggregation in the absence of any other agonist. The most active agonist from the glyceryl-ether glycolipids was almost nine orders of magnitude less potent than PAF, which means that its action through PAF-R would minimize PAF effects and finally could act as a PAF inhibitor. Indeed, these glycolipids inhibited PAF- and THR- induced aggregation, with IC_50_ values ranging from 6 to 12 µL (expressed as initial volume of OO). The representative structure of OO glyceryl-ether glycolipids is shown in Figure 4 [42].

Since atherogenesis is clearly linked to endothelial cell function, other in vitro experiments have been performed with these types of cells that are commonly stimulated with lipopolysaccharides (LPS) or cytokines and the expression of adhesion molecules as well as the levels of soluble endothelial markers in blood are evaluated. It has been demonstrated that oleic acid at a concentration range from 25–100 μM decreased the stimulated expression of vascular cell adhesion protein 1 (VCAM-1) as well as the release of the macrophage colony-stimulating factor (M-CSF). Vitamin E (α-T, 25 μM) was reported to block the activation of NF-kB mediated by the 18-carbon fatty acids except oleic acid [58,59] and also at higher concentration (50 μM), inhibited the stimulated expression of VCAM-1 by less than 50% [60]. In addition, α-T at even higher concentrations (200 μM), reduced expression of intercellular adhesion molecule-1 (ICAM-1) and VCAM-1 on interleukin-1β (IL-1β)-stimulated endothelial cells as well as PAF-induced expression of the adhesion molecules, CD11b and CD18 on neutrophils [61]. Lastly, vitamin E, at final concentration 10^−4^ M, inhibited the activation of the nuclear factor kappa-light-chain-enhancer of activated B cells (NF-κB) induced by PAF as well as the expression of tumor necrosis factor alpha (TNF-α) induced by NF-κB in a macrophage cell line [62].

Among the phenolic compounds tested, it has been reported that only oleuropein decreased the stimulated expression of VCAM-1 and reduced the expression of E-selectin and ICAM-1 in endothelial cells with an IC_50_ value at approximately 15–20 μM [63]. In stimulated whole blood cultures, oleuropein glycoside at 10^−4^ M decreased the IL-1β concentration while it had no effect on IL-6 or TNF-α levels [64]. In addition, oleuropein (10^−4^ M) has been shown to increase NO production via inducible nitric oxide synthase (iNOS) activation from LPS-stimulated mouse macrophages [65]. It seems that the antioxidant activities and the scavenging potencies of the phenolic compounds resulting in decreased LDL susceptibility to oxidation are the ones mainly contribute to the endothelial function since strong associations between oxidative stress and impaired endothelial function have been extensively reported. Lastly, our group has shown that T reduced PAF production (IC_50_ value at 304 μM) as well as the activity of PAF biosynthetic enzymes (Lyso-PAF-AT, PAF-CPT) with IC_50_ values at 48 and 246 μM, respectively, in IL-1β-induced monocytes [66]. Other flavonoids that are not present in OO have been reported to strongly inhibit AT activity and activate TA in H_2_O_2_-treated endothelial cells [29].

### 3.2. Animal Studies

In the animal studies, rabbits, apolipoprotein E knockout (ApoE-deficient) mice, and rats have been frequently used as experimental models. Additionally, a variety of cardiovascular risk factors are evaluated, such as lipid profile, markers of oxidative stress and inflammation along with markers of platelet activation and endothelium function, while histological analysis of aortic atherosclerotic lesions are occasionally included in the research outcomes. In addition, most of the animal studies evaluating cardiovascular effects of OO microconstituents have been performed with the administration of an enriched OO rather than with the purified microconstituent. Therefore, we only present data from animal models that the function of discrete OO microconstituents can be concluded, emphasizing on the atherosclerosis-induced experimental models. It is noted that the majority of the studies conclude that supplementation of the atherogenic diet, with OO or VOO, reduces vascular thrombogenicity and platelet activation in animals or stops the progression of atherogenesis, but nonsignificant reductions have also been observed [38,67,68,69,70].

As far as the NL are concerned, supplementation (10%, *w*/*w*) with refined OO in ApoE-deficient mice diet, in other words with only the NL OO fraction, resulted in more atherosclerotic lesions, while the enrichment of the above refined OO in linoleic, phytosterols, tocopherols, triterpenes, and waxes delayed the development of atherosclerosis [71]. Additionally, a dietary supplementation (corresponding to 15%, *w*/*w* OO) using the NL fraction extracted from OO, in rabbits fed with a cholesterol-rich diet, resulted in no significant reduction [72].

Three animal studies with genetically modified mouse models have examined the effect of plant sterols and stanols supplementation, on the development of atherosclerotic lesions. The first one was performed with male ApoE-deficient mice fed with a Western-type diet (9%, *w*/*w* fat and 0.15%, *w*/*w* cholesterol) for 18 weeks, while plant sterols supplementation (2%, *w*/*w*) for the subsequent 25 weeks did not provide any significant change in lesion development [73]. Contradictory results have been reported in the other study with the same animal model and type of atherogenic diet. In the latter, male ApoE-deficient mice fed for 6 months with either the atherogenic diet or the diet supplemented with 2% mixed plant sterol ester mostly containing sitosterol (46.2%), campesterol (25.3%), and stigmasterol (19.1%) resulted in reduced atherosclerotic lesion formation [74]. In the third study, the effect of 2% supplementation for 12 weeks with either plant sterols (campesterol and sitosterol) or stanols (campestanol and sitostanol) or atorvastatin or their combination was investigated in LDL-receptor deficient (LDLr+/−) mice that were prior fed with a Western-type diet for 33 weeks. In this study, male mice did not develop any detectable lesion formation after the consumption of the same Western-type diet despite the fact that their serum total cholesterol concentration was 2 mmol/l higher compared to the female ones. A synergistic effect of atorvastatin plus plant sterols or stanols in decreased serum cholesterol was reported, and no effects on early and regular fatty streaks were observed but a reduction in mild (characterized from presence of a fibrous cap and foam cells located in the media) to severe plaque formation was demonstrated in female mice [75].

The effect of vitamin E and especially γT and its metabolites have been only examined in inflammation-induced animal models and it was reported that its administration at concentration 100 mg/kg decreased inflammatory markers, such as prostaglandin E_2_ (PGE_2_), LTB_4_, and TNF-α [46,76].

Two studies have been performed with squalene, one in ApoE-deficient male and female mice with a dose of 1 g/kg/day, which showed a reduction in atherosclerotic plaques only in male animals [77], while the other one, with 3% dietary enrichment in rabbits, resulted in an increase in liver weight and liver nonsaponifiable compounds without affecting atheromatic plaques development [78].

The capacity of the individual phenolic compounds to inhibit in vitro platelet aggregation is observed only with HT in animal studies. Specifically, rats were fed with more than 50 mg/kg/day HT in order to achieve 50% inhibition of COL-induced aggregation [79,80]. The results concerning the anti-atherogenic action of HT are contradictory, since administration of 10 mg/kg/day HT in ApoE-deficient mice increased atherosclerotic lesion and monocyte activation [81], while significant reduction of the atherosclerotic lesions has been reported after supplementation with 4 mg of HT/kg in hyperlipidemic rabbits [69]. Animal studies with oleuropein have been performed in inflammation-induced models, where the administration of 10–500 mg/kg/day decreased inflammatory markers and symptoms [82]. Additionally, 20 mg/kg/day of oleuropein exerted protective effects against myocardial ischemia/reperfusion in rats and in cholesterol-fed rabbits [83,84] and prevented the development of myocarditis in rats [85].

Additionally, an intervention performed in ApoE-deficient mice fed with a high fat diet, supplemented (15%, *w*/*w*) with either an EVOO or an EVOO containing four times higher amounts of phenolic compounds, showed that both groups equally downregulated expression of the adhesion molecules, TNFα and NF-kB, and exhibited a reduction of macrophage infiltration in a similar extent, even though only the enriched EVOO enhanced the endothelium-dependent vasodilation in aortic rings [86]. Another study with a very similar design in LDL-receptor knockout Leiden mice concluded that only EVOO protects against high fat diet-induced atherosclerosis development in the context of obesity, while the EVOO that was enriched with phenolic compounds did not present an antiatherogenic effect [87]. Finally, supplementation of a high-cholesterol diet (HCD) with 10% either EVOO or phenolics deprived-EVOO in rats concluded that both supplementations decreased E-selectin and VCAM-1 compared to cholesterol fed rats, while no difference between the two EVOO groups was detected. No differences in the formation of atheromatous plaques were detected among groups since experimental rats do not develop lesions unless fed on specifically designed diets [88].

The above data clearly demonstrate that the higher amounts of phenolic compounds in enriched OOs do not improve the anti-atherogenic effect of EVOO and also reveal the existence of other microconstituents in the PL fraction of OO, which contribute to its protective action.

Indeed, diet supplementation (15%, *w*/*w*) with either OO or OO polar lipid extract (OOPLE), which included the PAF-inhibitors identified as glyceryl-ether glycolipids, in rabbits fed with a cholesterol-enriched diet, resulted in almost equal reduction in lesion thickness and retainment of vessel walls elasticity compared to the hyperlipidemic animals. It is noted that the OOPLE contained negligible amounts of phenolic compounds (0.030 μmol phenolics expressed as gallic acid /mg polar lipids). The protective effects of both diets were accompanied with an attenuation of ex vivo platelet aggregation as well as of plasma oxidation. Among all biochemical parameters evaluated, only the decreased PAF-induced platelet aggregability was correlated with the reduction of the atherosclerotic lesions [72]. The efficiency of PAF inhibitors and especially PAF-R antagonists to protect against atherosclerosis development was firstly documented with the administration of BN 52021 (ginkgolide B) at a dose of 20 mg/kg in rabbits fed with a cholesterol-enriched diet [89] and has also been reported with the fraction of polar glyco-lipids isolated from fish, leading to lower PAF levels and its activity in the blood of rabbits, by down-regulating PAF biosynthesis and up-regulating PAF catabolism [90]. Additionally, intravenous administration of adenovirus-mediated expression of PAF-AH in ApoE–deficient mice inhibited atherosclerosis by reducing oxidized lipoprotein accumulation [91].

### 3.3. Human Studies

The benefits for cardiovascular health in humans from the consumption of OO are established both by the first randomized controlled trial, The PREDIMED study, and subsequently by intervention studies and meta-analyses. They all show that OO consumption results in a reduction of CVD risk, while its inverse association with myocardial infraction or death from CVDs remains a controversy [92].

As far as the MUFAs are concerned, a meta-analysis of 32 cohort studies showed that MUFAs of mixed animal and plant origin did not result in any significant effects with respect to a reduced risk of all-cause mortality, CVD events, and stroke. In contrary, significant correlations were only observed with high intake of OO, but it should be noted that the specific sources of MUFA have not been indicated in every study and also that their predominant source significantly vary among studies [40].

Phytosterols are well recognized for their cholesterol-lowering efficacy since a great number of clinical trials have demonstrated that intake of 2 g phytosterols per day leads to lower LDL-C approximately by 10%. A recent review that presents the intervention studies with phytosterols concluded that their supplementation did not result in any beneficial change in the markers of cardiovascular risk events, including parameters of the vascular function [93]. This conclusion is in line with other published studies that support the idea that even though cholesterol levels are statistically correlated to CVDs, it is not an important causal factor for atherosclerosis and they also suggest that other alternative hypotheses for the etiology of CVDs should be formulated [94,95].

The majority of human studies focus on vitamin E and phenolic compounds. As far as vitamin E is concerned, the average daily recommended amount (RDA) for adults according to the National Institutes for health (NIH) as well as to the European Food Safety Authority is no more than 15 mg [96,97]. The amounts of vitamin E supplementation among studies far exceed the RDA, varying from 268–536 mg/day in the case of natural vitamin E (the naturally occurring αT stereoisomer), and from 50–300 mg/day of synthetic vitamin E (racemic mixture of all stereoisomers of αT). Mixed tocopherols (100 mg γT, 40 mg δT, and 20 mg αT) supplementation in healthy subjects slightly decreased ADP-induced platelet aggregation (approximately 10%), while αT alone had no effect. Both supplementations increased nitric oxide release and superoxide dismutase protein (SOD) in platelets [98]. Additionally, dietary supplementation with D-αT (1500 IU/day for 14 days) in males had no significant effect in the aggregation induced by ADP or COL either performed in WB or in PRP, in plasma phospholipase A_2_ activity, or in plasma lyso-PAF levels [99]. Randomized clinical trials doubt the efficacy of vitamin E supplements to prevent CVDs and the studies examining the association of serum vitamin E with the risk of CVD show inconsistent results [100]. Additionally, meta-analyses have reported a significantly reduced risk for ischemic stroke and increased risk for hemorrhagic stroke even though the absolute effects were small [101] and also that high-dosage of vitamin E supplements (≥400 IU/d) may increase all-cause mortality [102]. Lastly, a two-sample mendelian randomization study concluded that each 1 mg/L increase in vitamin E is significantly associated with coronary artery disease (CAD) and worse lipid profile [100]. Although the protective role of vitamin E on lipid peroxidation in vitro has been confirmed in many studies, it has not yet been demonstrated whether it exhibits the same effect in humans.

On the other hand, human studies have shown that plasma antioxidant activity increases after the intake of OO phenolic compounds and that an inverse relationship between LDL oxidation and phenolic concentration exists [103]. The efficiency of any individual OO phenolic compound to favorably modulate platelet functionality in vivo cannot be reliably drawn by the existing human studies since all the intervention groups consumed EVOO, which was different only in the amount of the total phenolics, while in some cases, the interventions did not even have similar FA profile [50]. A recent meta-analysis of 26 RCTs concluded that OO with high content in polyphenols (over 150 mg per kg of oil) can improve outcomes related to total and HDL cholesterol and oxidative stress markers when compared to low polyphenol OO [104].

Clinical trials with the use of a single phenolic have not been performed so far, with the exception of HT, which has been tested in three studies. In a cross-over, placebo-controlled trial, intake of 15 mg/day of HT for 3 weeks resulted in an increase of thiols and of total antioxidant status while malondialdehyde levels were reduced, ox-LDL concentration was unaffected and the expression of SOD1 was the only one increased among a variety of genes belonging to inflammatory and oxidative stress pathways that have been evaluated [105]. In a previous trial with healthy subjects, consumption for 1 week of olive mill wastewater extract fortified with two different doses of HT (5 and 25 mg/day) failed to induce any significant effect on a variety of markers of cardiovascular disease, such as lipid profile, inflammation, and oxidation markers [106]. Similarly, when HT was administered at a daily dosage of 45 mg for 8 weeks to subjects with mild hyperlipidemia, no influence in markers of cardiovascular disease, such as blood lipids, inflammatory markers, liver or kidney functions were detected [107]. In a crossover, randomized, double-blind study, postprandial ox-LDL concentrations decreased at 0.5 h and 4 h after the intake of 5.25 mg HT in a form of HT-enriched biscuits [108]. Additionally, healthy subjects who had consumed 20 mg oleuropein just before lunch presented an improved postprandial glycemic profile and decreased production of 8-iso-PGF_2α_, while NADPH oxidase 2 activation was down-regulated at 2 h after lunch consumption [109].

The daily consumption of olive leaf extracts, containing 50–136 mg oleuropein, along with 6–10 mg HT and other phenolics (less than 1 mg each), has been evaluated in one acute intervention trial on healthy volunteers and in three randomized, controlled studies for 6 to 12 weeks. In general, the results showed a positive effect on systolic and diastolic blood pressure, while no significant action was recorded on the production of inflammatory cytokines and other markers [110,111,112,113].

The lack of significant effects in the clinical trials with phenolics is usually attributed to their low bioavailability and low plasma or tissue concentration, especially for polyphenols or to their chemical modifications by the intestinal microbiome. Relative studies have documented that OO phenols are absorbed, and also that after the intake of phenolic compounds either as EVOO, as virgin OO, as olive leaf extracts, as pure HT, or as olive by-products or precursors, HT metabolites are the ones predominantly circulating in the blood [114,115].

On the other hand, the effectiveness of the use of antioxidants in atherosclerosis is still questionable while skepticism exists for their probable pro-oxidative action [116]. It is no coincidence that the overexpression of SOD not only has no effect on atherosclerosis but is also positively associated with the lesion area in transgenic mice [117].

## 4. Olive Oil By-Products

Olive pomace (OP) is the main by-product of the OO production process. It is a semi-solid residue and it consists of pieces of skin, pulp, olive kernel, and some oil. Even though the type and the amount of OO b-product depends on the type of the extraction method, namely press method, two-phase, and three-phase system, it has serious environmental impact. The two-phase OO extraction process, which is characterized as ecological compared to the three-phase system, produces OP with a moisture content of 65%, also named “*alperujo*”, which accounts for approximately 80% of the olive weight.

OP contains carbohydrates mainly in the form of polysaccharides, lipids, phenolic compounds as well as inorganic compounds. It contains the same classes of phenolic compounds with the OO and the predominant ones are T, HT, and p-coumaric acid [118]. In addition, the fraction of OP PL includes PAF-inhibitors having the structure of glyceryl-ether glycolipids sharing similar biological activity and structurally characteristics to the ones from OO PL [119]. Besides its toxicity, OP is a source of valuable compounds, so a variety of methods—some of them as patents—have been developed in order to purify them and specially to obtain phenolic compounds. Their suggested applications range from nutritional supplements and functional foods to animal feed.

As mentioned above, the triterpenes, erythrodiol, uvaol, oleanolic, and maslinic acids, are present in detectable amounts in OP oil (OPO). These triterpenes are described mostly in in vitro studies but also in ApoE mice and rats, to present anti-inflammatory, anti-oxidant properties, vasodilatory properties mainly mediated by the endothelial production of NO as well as cardioprotective effects. The efficient doses range from 10 to 400 μM in the in vitro experiments while 50–100 mg/kg/day are supplemented in the feed of the experimental animals [120]. In a randomized, crossover trial, healthy adults consumed for three weeks, a daily dose (30 mL) of three OOs, namely a standard VOO (124 ppm of phenolic compounds and 86 ppm of triterpenes) as a control, a VOO fortified with phenolic compounds (490 ppm of phenolic compounds and 86 ppm of triterpenes), and another VOO fortified with both phenolic compounds and triterpenes (487 ppm of phenolic compounds and 389 ppm of triterpenes). The results demonstrated that the VOO with the highest triterpenes amount had no additionally effect in biomarkers related to metabolic syndrome and endothelial function [121], while its consumption resulted in decreased DNA oxidation and plasma IL-8 and TNFα concentrations compared to the other two oils [122]. These results are also confirmed by a randomized, blind, crossover, controlled, clinical trial, in healthy and hypercholesterolemic participants that consumed daily 45 g of either OPO or high-oleic acid sunflower (HOSO) oil for 4 weeks. Both oils had comparable oleic acid content while tocopherols and sterols were higher in HOSO and OPO and showed a higher content of aliphatic alcohols, squalene, and triterpenes. The intake of OPO resulted in lowered LDL-cholesterol and Apo B levels compared to the HOSO but without affecting blood pressure, endothelial function as well as a significant number of inflammatory biomarkers [123].

Our research group has demonstrated that the PL fraction from the olive pomace (OOPLE) inhibited in vitro PAF-induced platelet aggregation with an IC_50_ value equal to 1.1 × 10^−10^ M, expressed as sugars, and equivalent to the corresponding one induced by the PL from OO (IC_50_ PAF 1.5 × 10^−10^ M, expressed as sugars). This inhibition is attributed to binding to the PAF-R with EC_50_ value at 0.42 × 10^−7^ M (expressed as sugars) which is comparable to the one of the well-known PAF-R inhibitor, BN 52021 (EC_50_ = 2.3 × 10^−7^ M), a ginkgolide of the Chinese plant *Gingko biloba*. In addition, the OOPLE had the ability not only to inhibit the formation of atheromatous plaques in hypercholesterolemic rabbits, but also to improve artery elasticity and to reduce the thickness of the already formed atheromatous plaques comparable to simvastatin, which was used as a positive control (Figure 5) [124]. In order to investigate the effect of the OOPLE in humans, a low-fat yogurt was enriched by 0.4%, *w*/*w* with this fraction. The randomized, three-arm, double-blind, placebo-controlled, parallel-group trial, performed in apparently healthy and mainly overweight adults, showed that the daily intake of the yogurt supplemented with OOPLE (150 g) for a period of eight weeks resulted in lower levels of inflammatory cytokines and in reduced platelet sensitivity against PAF, compared to plain yogurt, while no impact on energy intake, body weight, glucose, and lipid metabolism was detected [125]. Additionally, the intake of the enriched yogurt favorably modulated PAF metabolism by reducing PAF-CPT and LpPLA_2_ activities [126]. In addition, diet supplementation with exactly the same enriched yogurt (2%, *w*/*w*) in rabbits fed with a cholesterol-enriched diet resulted in a decrease in the maximum and average thickness of atheromatous plaques by 54% and 57%, respectively [127]. Finally, preliminary results from another intervention study showed that the consumption for 4 weeks of fillets from fish fed with a diet, where fish oil/meal in the grow-out feed of gilthead sea bream was partly replaced with the OOPLE, led to reduction of platelet sensitivity against PAF, and ADP by 44% and 67%, respectively, compared to consumption of fillets from fish fed the typical diet [128]. The reduced platelet sensitivity observed in both trials after the intake of the OOPLE rich in PAF antagonists, is in accordance with the results from our previous intervention studies where the same effect was recorded in healthy subjects and type II diabetic patients after one-month consumption of Mediterranean-type foods rich in PAF inhibitors [129,130].

Apart from OP, two human trials have been performed with the olive mill wastewater (OMWW). In the first one that was a non-placebo postprandial intervention, healthy subjects consumed 2 mL of a commercially available OMWW preparation containing approximately 50 mg of HT and 8 mg of oleuropein along with other phenolic compounds. The results showed that the glutathione plasma concentration increased, while no difference in plasma antioxidant capacity was detected [131]. In the second study, five type I diabetic patients received 25 mg of HT the first day and 12.5 mg/day the following three days as a HT-rich phenolic extract from OMWW along with breakfast. The results demonstrated a significant decrease in the serum TXB_2_ production after blood clotting at the end of the intervention [132].

## 5. Conclusions

OO microconstituents exert in vitro anti-thrombotic, anti-inflammatory, and anti-oxidant activity. MUFA, phytosterols, squalene, and triterpenes do not show promising results against atherogenesis when tested separately in in vivo studies. Randomized clinical trials doubt the efficacy of vitamin E supplements to either present an anti-thrombotic effect or to prevent CVDs. As far as phenolic compounds are concerned, the data reveal that HT and oleuropein, either supplemented separately or in combination, do not induce any significant effect on a variety of markers of CVDs in humans. Additionally, further enrichment of OO with these phenolics does not result in additional anti-atherogenic effect. On the other hand, PL from OO and OO by-products that act as PAF inhibitors inhibit the formation of atheromatous plaques in hypercholesterolemic rabbits, and also ameliorate inflammation and reduce platelet sensitivity in healthy adults. In conclusion, VOO shows anti-atherogenic effects, which can be attributed to the synergistic action of its microconstituents, mainly PL that act as PAF inhibitors, tyrosol and oleuropein as well as αT that also exert anti-PAF activity. Finally, as the EFSA suggests the daily intake of moderate amounts of OO that can be easily consumed in the context of a balanced diet is considered satisfactory for the health of the general population [133].

## Figures and Tables

**Figure 1 biomolecules-13-00700-f001:**
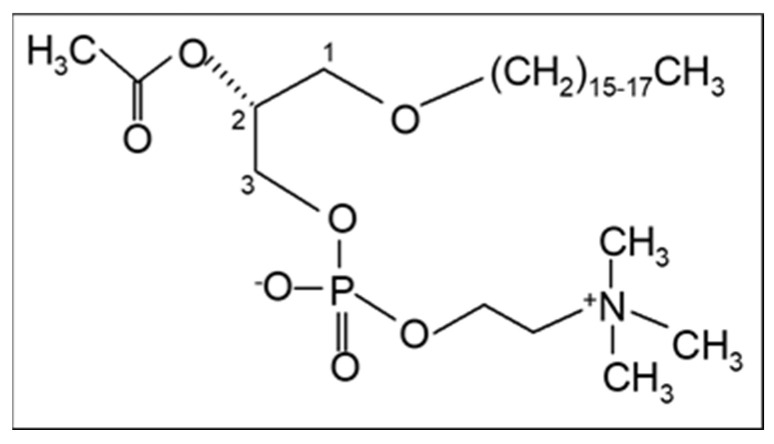
The chemical structure of Platelet-Activating Factor, PAF (1-O-alkyl-2-acetyl-*sn*-glycero-3-phosphocholine).

**Figure 2 biomolecules-13-00700-f002:**
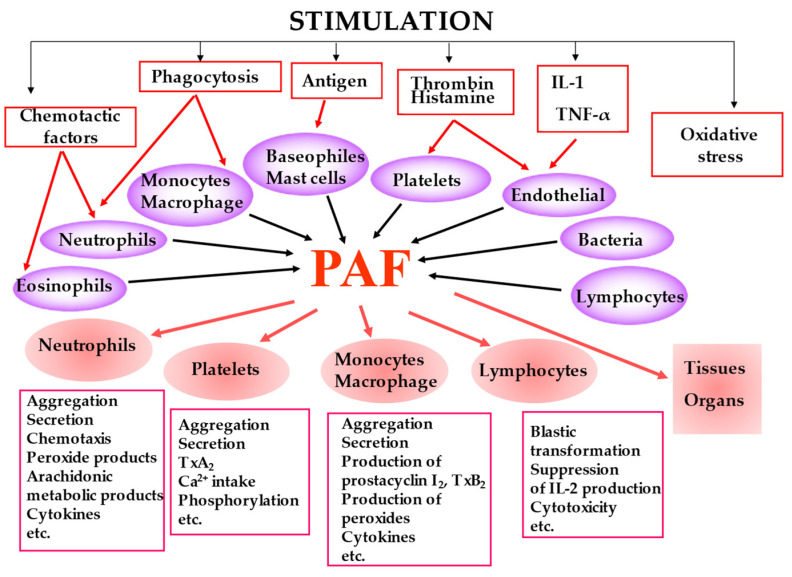
The main cell types that produce PAF under stimuli along with the key biological responses of PAF.

**Figure 3 biomolecules-13-00700-f003:**
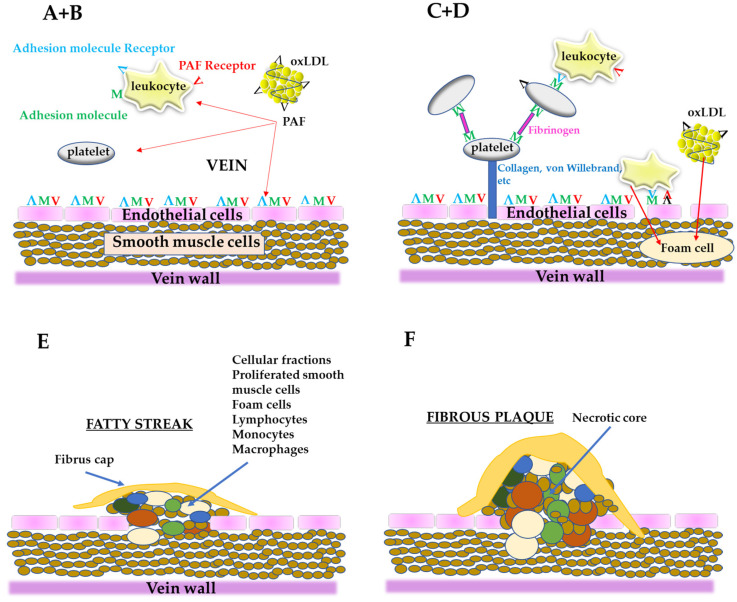
A simplified schematic representation of the stages of the formation of atheromatic plaque according to “the PAF implicated atherosclerosis theory”; (**A**) PAF and PAF-like activity lipids generated from oxLDL or from other sources can activate endothelial cells, leukocytes, and platelets; (**B**) Endothelium is activated (dysfunctional endothelium) along with the activation of leukocytes and platelets; (**C**) After platelet adhesion, through fibrinogen, the cells are stabilized with collagen, von Willebrand factor, etc., and make along with leukocytes a layer on the dysfunctional endothelium; Leukocytes roll along the layer of platelets-leukocytes or along the dysfunctional endothelium; When leukocytes are bound to the dysfunctional endothelium through adhesion molecules and PAF, arrest rolling; (**D**) Leukocytes move into sub endothelium space (migration) and differentiate into macrophages; Dysfunctional endothelium also permits oxLDL to migrate to the sub endothelium and is up taken by macrophages to form foam cells; (**E**) The proliferation by PAF of smooth muscle cells takes place, leading to corrupted sub endothelial architecture; The foam cells accumulate, the intimal thickness increase, and fatty steak is formed; (**F**) Within this localized lesion, continued inflammation can lead to cellular necrosis and further recruitment of monocytes and lymphocytes, with a concomitant release of different substances; This can set the stage for focal necrosis within the lesion and its autocatalytic expansion; The so-called fibrous plaque is formed.

**Figure 4 biomolecules-13-00700-f004:**
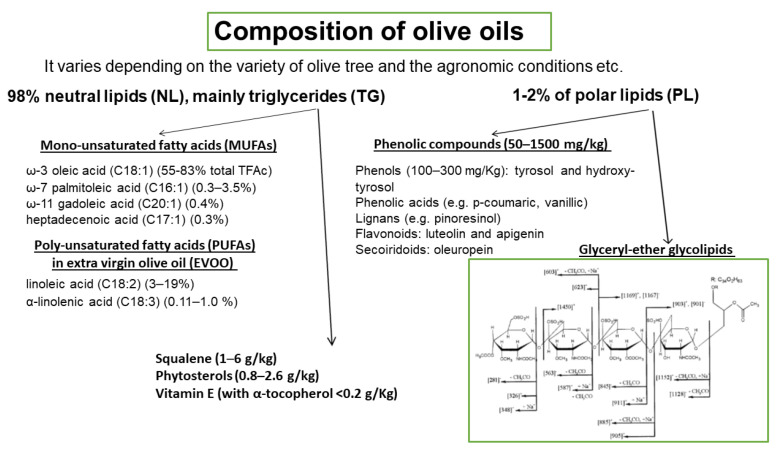
Composition of olive oils. Representative structure of olive oil glyceryl-ether glycolipids.

**Figure 5 biomolecules-13-00700-f005:**
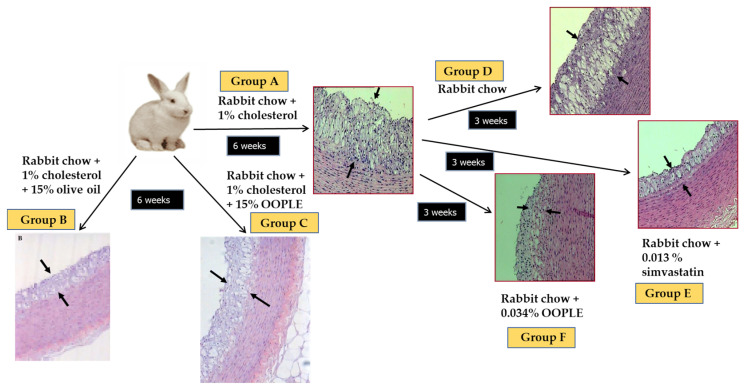
Experimental protocol and representative microphotographs of atheromatous lesions of aortic wall sections obtained from animals. The arrows indicate the observed atheromatous lesions. Group A: 6 weeks atherogenic diet; group B: 6 weeks atherogenic diet enriched with 15% OO; group C: 6 weeks atherogenic diet enriched with 15% OOPLE; D: 6 weeks atherogenic diet and additional 3 weeks normal diet; group E: 6 weeks atherogenic diet and additional 3 weeks normal diet plus simvastatin; group F: 6 weeks atherogenic diet and additional 3 weeks normal diet plus OOPLE.

## Data Availability

Not applicable.

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
