# Peer review of "Protective Effect of Olive Oil Microconstituents in Atherosclerosis: Emphasis on PAF Implicated Atherosclerosis Theory"

_biomolecules, 2023, doi:10.3390/biom13040700_

Round 1

Reviewer 1 Report

The manuscript by Antonopoulou and Demopoulos, outlines beneficial aspects of micro-nutrients of olive oil in PAF-mediated pathology, particularly in atherosclerosis. 

The manuscript is generally well written but has potential for improving at many places:

a) Extensive correction of language is recommended. For example: Fig.3 white blood cell is spelled as leucocytes in few places and leukocytes at other places. Line number 382 - vasodilation is misspelled. Line number 312- sentence is grammatically wrong. In the same page, sentence 337- I think it is "fed with" not just "fed". Such mistakes have been found throughout the manuscript. (check line 71-72, 145-150, 165-168, 171-173, 188, 196, 205-206, 218-220, 249-254, 265, 273-275, 278-281, 295-297, 312-313, 348, 366-367, 422, 431-432, 434, 437-439, 467-468 - does not make any sense. Line 478- was should be replaced by were. 503, 508, 519 (italics), 521-525. 555-558 - reframe the sentence. reframe the sentence 584.

b) Uniform citation of the bibliography is needed. Check reference 11, 14, 26, 82 and many more have to be corrected.

c) Technical issue: sentence 66-67, if it is not an ester bond but an ether bond, then sn-1 residue will be a fatty alcohol.

d) A general observation - the beneficial effects of these micronutrients occurs at micro-molar level, in isolated system. Therefore, really whether these micronutrients contribute to beneficial effects of olive oil, is a big question mark ( in other words, how much olive oil should we consume?). 

e) Abbreviated terms needs to be expanded at certain places for reader’s benefit.

Reviewer 2 Report

The present review focusses on the effect of olive oil microcomponents in the different mechanisms that initiate and develop atherosclerosis, particularly related to the PAF theory. The review is extensive and covers in vitro, animal, and human studies where the effects of olive oil microcomponents or olive oil itself have been studied.

Overall, I recommend publication after revision of the following minor comments:

- Review the abbreviations used along the text (OO, PAF, CVDs, ox-LDL, ADP, THR,…). Introduce them for the first time used on the text and then make use of them.

- I suggest using microcomponents or microconstituents in the text, but not both.

- Since the postprandial effect of olive oil and its microconstituents have been tested postprandially, I think that some references should be included in the Human studies section to take into account that the described effects are also observed inmediately after their ingestion.

Round 2

Reviewer 1 Report

Although the review manuscript submitted by Demopoulos et al has considerably improved, authors have not adequately corrected the manuscript both for language and grammar. The required corrections have been attached in the file below. 
